# Quality Study on Vehicle Heat Ventilation and Air Conditioning Failure

Dina Diga [1], Irina Severin [1,*] and Nicoleta Daniela Ignat [2]

[1] Faculty of Industrial Engineering and Robotics, University Politehnica of Bucharest, 060042 Bucharest, Romania; banteadina@gmail.com
[2] Faculty of Entrepreneurship, Business Engineering and Management, University Politehnica of Bucharest, 060042 Bucharest, Romania; nicoleta.ignat@upb.ro
* Correspondence: irina.severin@upb.ro; Tel.: +40-726-283-850

**Abstract:** The automotive industry has set a highly demanding standard to meet customer satisfaction. The paper aimed to detail how quality analysis has been conducted to state the main causes that generated nonconformities of heat, ventilation and air conditioning (HVAC) systems. Problems have been reported on HVAC systems, such as noise, not being cold enough and moldy smell issues. All three problems determined by customer reports initiated the first contribution of this paper, namely by an initial quality study, and generated the investigation using is/is not problem scoping, data analysis, and graphical analysis. Pareto analysis and the Plan, Do, Check and Act (PDCA) approach are used to highlight the traceability of the actions performed in the evaluation of the problems and the detection of the causes related to each problem. The data analysis process and the data obtained from the analysis are the core of this paper. An immediate action plan is proposed, concluding with the hypothesis that the root cause is the blockage of the heater. This methodology has significant potential for being implemented, even for other components in the same industry or different sectors.

**Keywords:** HVAC system; failure analysis; market study; customer verbatim; quality tools





## 1. Introduction

Automotive companies producing heating, ventilation and air conditioning (HVAC) systems try to exceed the technical limits to meet the needs of their customers by streamlining by reducing weight, size and costs. At the same time, demanding better-quality results than previous ones is often impossible to achieve. The identification of problems, their definition, and prognosis have attracted increased attention from engineers and researchers in terms of HVAC in the context of improving the comfort offered to the customer.

The purpose of determining customer satisfaction is to meet their needs, focusing on the satisfaction of demanding customers and the desired level of performance. The tasks of a quality control system are to identify, analyze and control, based on customer reports, the main failure modes that lead to the root causes and generated problems in HVAC systems.

In recent years, engineering and technological advances have progressively improved automotive thermal management. Motivational factors, the characteristics of new vehicles and their small size, the increase in the number of electric vehicles, the concern of consumers regarding fuel consumption, as well as consumer demands and political impacts, have contributed to the importance of this field [1]. The components of a ventilation system include mechanical and electrical elements. Being subjected to stress, there is the possibility of reducing their life cycles, and the reliability of the system must be evaluated in order to establish and schedule specific maintenance and repair activities [2]. Customers' expectations for increased comfort and luxury always require continuous development of vehicle HVAC control systems [3].

With vibrations, an HVAC heating and ventilation system generates various sounds, such as whistling or humming. This type of noise can identify and determine the causes of

problems [4]. Several failure modes have been investigated by measuring and controlling the requirements of air flow, power, temperature, and pressure gradients in a vehicle's HVAC [5].

According to Vishwanadha, the most important function generated by an HVAC system is the thermal and acoustic comfort of a customer during a trip [6].

In his paper, Vranău indicates that the conformity of technical comfort is given by the controlled four-meters and measurements with the help of special equipment meeting acoustic, thermal, and visual specifications [7].

The automotive HVAC industry is aiming for exponential growth in terms of efficiency and measured quality levels; the goal being to reach the maximum operating thresholds in terms of low costs and market exposure [8].

A number of non-conformities and a lack of documentation related to HVAC systems have been identified in literature reviews, such as:

- Lack of a standards on how to measure the noise in the HVAC system of a vehicle;
- There is no clear evidence on noise, and it is not recorded in specifications of noise level of analysis;
- Laboratories deliver results based on limited data, with different tools, which are not standardized due to a lack of information;
- Existing data are difficult to compare and interpret, so automotive companies deliver absolute noise levels, measured in soundproof environments;
- The large number of manufacturers and models of components makes it difficult to compare products between brands [9].

This paper highlights the usefulness of quality evaluation techniques for the analysis of non-conformities in HVAC systems. The results of the present study can be used, not only to understand how a system fails, but also to improve the ventilation flow of an HVAC module [10] and to validate the predictions of a study.

## 2. State of the Art

The key objectives of an HVAC system—heating, ventilation and air conditioning—are to maintain good indoor air quality through adequate filtration ventilation and ensure thermal comfort. Vehicle manufacturers are working hard to develop HVAC systems capable of streamlining the level of comfort of drivers [11]. According to current standards, subsystems, such as the condenser, air cooler, oil cooler, or engine coolant, have their own cooling systems. Thus, the literature cites cooling configurations and system changes to the HVAC refrigeration cycle [12], which is an important point to consider in the present study.

The specialized literature states three levels in which the research and development of the air flow and air distribution in vehicles is carried out:

- The first level defines the optimal conditions for satisfying passengers through thermal comfort;
- Following that, the second level focuses on real results determined in the laboratory and their generation through measurements and reported experiments;
- A third level in development would be the validation of these experiments using computer simulations [13].

Using an HVAC system:

- Temperature can be raised or lowered;
- Humidity can be raised or lowered;
- Proper filtration of air can be maintained;
- Proper air movement;
- Outside air can be added and removed;
- Holding air contamination within acceptable limits.

Hypotheses of failure
Possible failure modes under operating conditions:

- Hot and cold;

- Dry and wet;
- Dusty and dirty.

  Possible failure modes under usage:

- Rough environment;
- Average life cycle—above;
- Average life cycle—below.

  Possible failure modes under incorrect assembly:

- Wrong components have been improperly replaced;
- The components have been assembled incorrectly;
- The components have been omitted [14].

The sustainability strategy [15] in this case involves the appropriate improvement of climatic conditions of a vehicle and the prevention of any risk in order to create a safer journey for passengers [16].

Revised structure of the literature presents important points of the failure of an HVAC system related to time and the need for a quality study that will highlight the need for continuous customer satisfaction.

In order to elucidate the customers' requirements regarding the noises and heat losses detected with reference in the HVAC system of its vehicles, a series of investigations was made through a quality study. In the study presented in the "Case Study "section, it is designed to define, analyze and solve problems starting from customer verbatims. This study describes the HVAC system and the effects measured using quality instruments. The investigations were developed and presented through data analysis and improvement actions.

The role of an HVAC system is to act as a climate-control system, thus managing the appropriate desired temperature of an environment. The refrigerant line is part of an HVAC system and is a closed loop system through which the refrigerant circulates to absorb heat from the air. The refrigerant line is connected to a compressor, coupled to the engine, and is mounted on the vehicle body in different locations, depending on the specific requirements according to the assembly procedure [3,17]. Figure 1 shows a detailed presentation of an HVAC system by presenting the essential components that contribute to the effective functionality of the system.

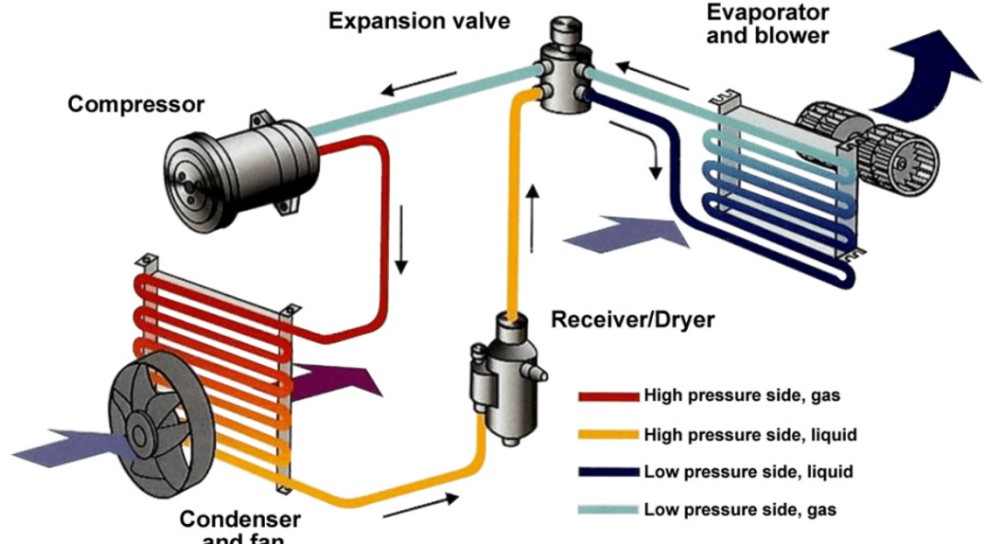

**Figure 1.** HVAC system (https://www.pngwing.com/en/free-png-yjfzx Accessed on 18 October 2021 9:57 P.M.).

The components of an HVAC system that contribute to the distribution of air flow and allow thermal conditioning to areas of the vehicle cabin are the receiver dryer, evaporator and blower, expansion valve, compressor and condenser, and fan.

A flowchart was developed and is shown in Figure 2 to represent HVAC conditions in terms of system design, process, and analysis. HVAC performance is measured based on customer expectations.

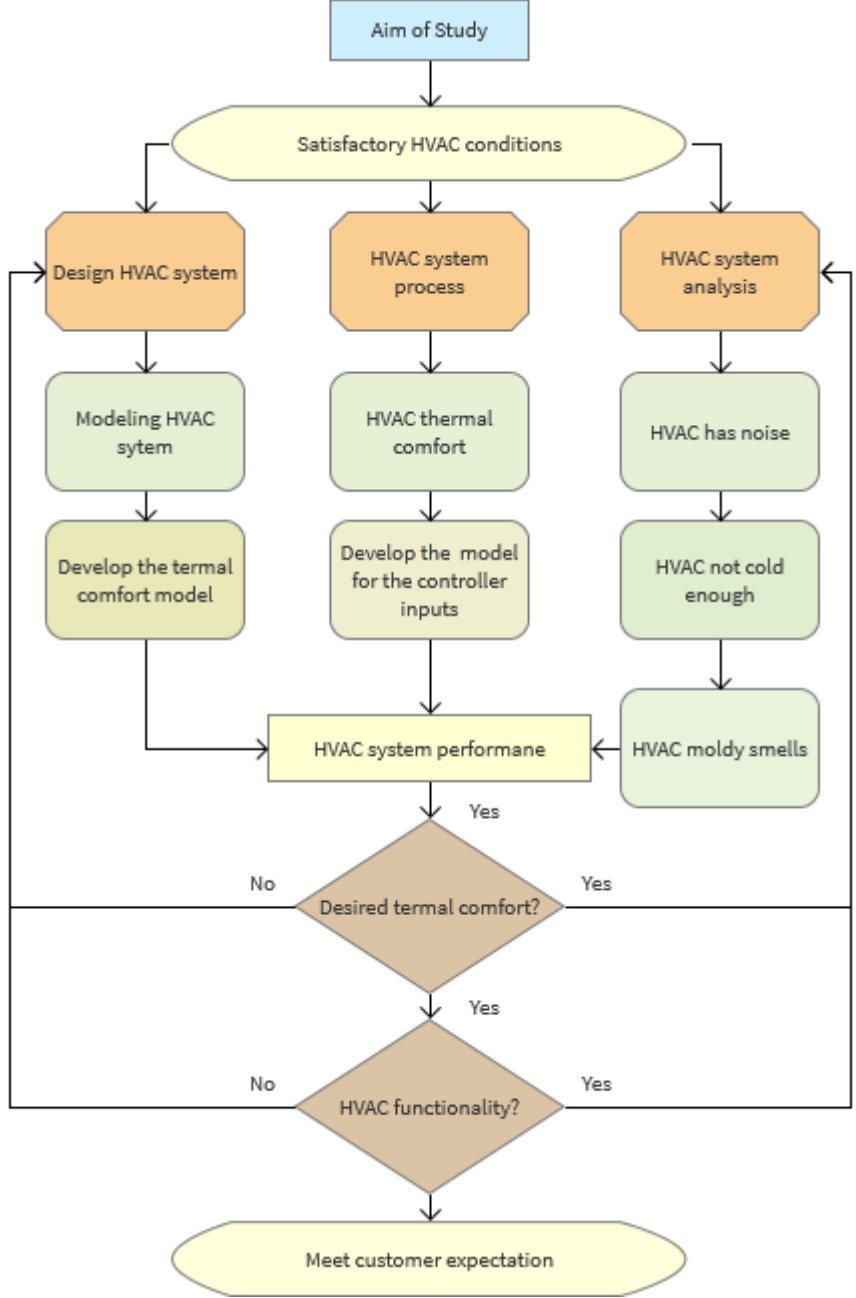

**Figure 2.** Flowchart for the HVAC system process.

The flowchart presents three possibilities that are desired to satisfy customer expectations. In this case, the conditions are met, and it is known that thermal comfort and HVAC functionality need to be improved. Otherwise, if these conditions are not met, the possibilities by which they would meet customer expectations have to be reassessed.

The developed process has the role of indicating problems detected in this study based on the analysis of defining and detecting the main causes of non-compliance.

### 3. Case Study

The study was conducted at an automotive company in the UK that produces HVAC systems. The data collected from the database were for the period of 2016–2021. The requirement of the study was to identify and evaluate problems reported by customers using quality evaluation techniques. The study involved the quality team, the HVAC engineering team, and the team responsible for worldwide customer complaints. The study was performed for the most common problems of an HVAC system, identified by engineers for different models of competitiveness vehicles, over a period of five years. The study aimed to help develop HVAC systems for the new-vehicle segment within the company.

The study was carried out following complaints reported by customers from several world markets regarding HVAC systems. The new vehicle segment saw a sharp increase of 81 customer concerns about the air conditioner, fan/blower noise, not cold enough, and moldy smell issues. A total of 50.3 customer dissatisfaction problems per 100 vehicles were found in the French, German and British markets.

Table 1 below shows these results; these scores are part problems per 100 vehicles in the French, German, and UK markets. A higher score means greater customer dissatisfaction.

**Table 1.** Initial quality study: HVAC customer issues.

| IQS No. | Customer Issues | Vehicle (pp/100) | Best Vehicle | Best Score | Segment Average | Gap to Best |
|---|---|---|---|---|---|---|
| 1 | Air Conditioner: Not Working Properly | 1.7 | Toyota Avensis | 0.3 | 1.2 | 1.4 |
| 2 | Air Conditioner: Noisy | 7.2 | Ford Mondeo | 2.2 | 4.0 | 5.0 |
| 3 | Air Conditioner: Not Cold Enough | 5.1 | VW Passat | 1.4 | 2.3 | 3.7 |
| 4 | Heater: Not Working Properly | 0.6 | VW Passat | 0.0 | 0.7 | 0.6 |
| 5 | Heater: Does not Get Hot Enough | 0.3 | VW Passat | 0.0 | 0.4 | 0.3 |
| 6 | Fan/Blower: Not Working Properly | 0.9 | Renault Laguna | 0.2 | 0.6 | 0.7 |
| 7 | Fan/Blower: Noisy | 11.7 | VW Passat | 2.8 | 3.8 | 8.9 |
| 8 | Front Defroster | 0.7 | Ford Mondeo | 0.2 | 0.6 | 0.5 |
| 9 | Rear Defroster | 0.1 | Renault Laguna | 0.1 | 0.2 | 0.0 |
| 10 | Heater/ Air Conditioning (AC) Smells Moldy/Stale | 2.3 | VW Passat | 0.9 | 1.2 | 1.4 |
| 11 | Windows Fog Up a Lot | 9.7 | VW Passat | 3.7 | 5.1 | 6.0 |
| 12 | Cannot Maintain Desired Temperature | 2.3 | Renault Laguna | 0.6 | 2.2 | 1.7 |
| 13 | HVAC Controls | 1.1 | Toyota Avensis | 0.8 | 1.6 | 0.4 |
| 14 | Other HVAC | 6.7 | Toyota Avensis | 0.2 | 1.9 | 6.5 |
| | Total | 50.3 | n/a | 13.2 | 25.8 | 37.1 |

As can be seen in the table, the study was performed on the problems in HVAC systems of several vehicle models, giving the best scores on the developed segment compared to the average segment scores. It can be seen that the VW Passat, Ford Mondeo and Toyota Avensis recorded the highest scores, which means the highest degree of customer dissatisfaction.

A comparison between the best score and segment average data can be seen in Figure 3 using a histogram representation. In the histogram, an obvious variation between the scores given by customers regarding the unfavorable impact and dissatisfaction generated by the appearance of these problems registered in the table in the customer verbatim section can be seen.

Starting from these records, the problems were investigated by defining the problem, starting from the customer complaints, and using the assigned symptoms and verification actions. All this evidence is presented in Table 2, where the impacts of these problems in terms of business and customer opinions are analyzed and defined.

At the same time, the 3Ws+H method was used to answer the most important questions involved in detecting problems, as shown in Table 3. This type of approach helps the quality and engineering teams to identify and estimate possible causes in the generation of these nonconformities. The problems are described by the answers received to the questions "What, Where, When and How", questions that facilitate better evidence and an interpretation of the existing information.

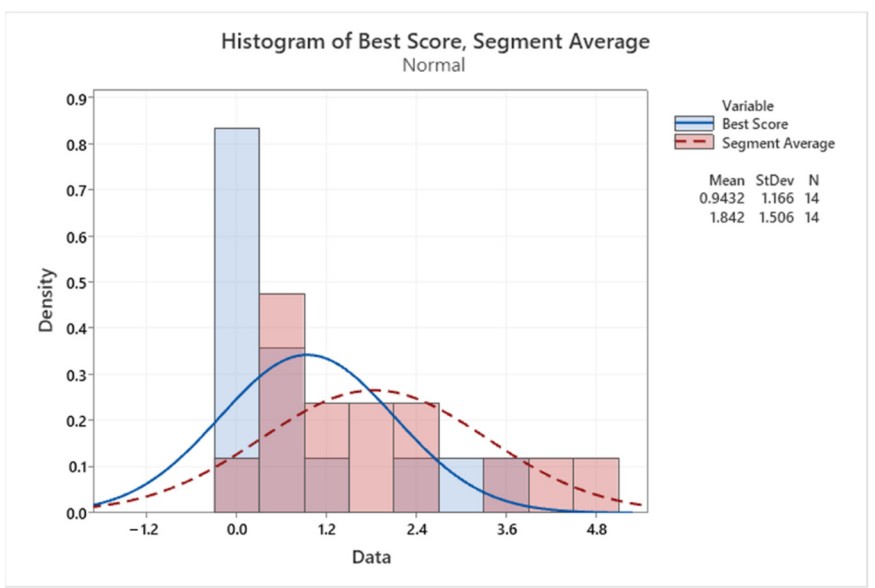

**Figure 3.** Best Score and Segment Average comparison.

**Table 2.** Initial quality study: problem investigation.

| Problem Definition | |
| --- | --- |
| Business Case | The new vehicle segment had poor results in the heat ventilation and air conditioning (HVAC) area |
| Symptoms | Air conditioner/fan/blower—noisy (1)<br>Air conditioner—does not get cold enough (2)<br>Heater/AC smells moldy/stale (3) |
| Voice of the Customer | The AC and fan are too noisy<br>Air Conditioner—does not get cold enough<br>Heater fan not working/AC smells |
| Customer Requirements | Heating, cooling and ventilation to work as required |
| Problem Scope | HVAC is noisy, not cold enough, and has moldy smell issues. Issues affect the new vehicle segment. |
| Actions | Check HVAC assembly issue; check AC compressor and Freon lines' pressure. Interior temperature sensor.<br>Check if the fan is working properly. |
| Project Requirement | 81 problems verbatim<br>Increased customer satisfaction/confidence |
| Resources Required | Body quality, supplier test facility, testing laboratories |

**Table 3.** Initial quality study: is/is not problem scoping.

| Problem Statement: HVAC Is Noisy, Not Cold Enough and Has Moldy Smell Issues | | | |
| --- | --- | --- | --- |
| 3Ws+H | Problem Description | IS | IS NOT |
| WHAT | What object<br>What defect | Toyota Avensis, Ford Mondeo, VW Passat, Renault Laguna<br>Too cold/too hot | Other car brands<br>There are no deformed components |
| WHERE | Where on object<br>Where seen since | Air conditioner/fan/blower not operating, noise and smell<br>France, Germany, UK | Other European countries |
| WHEN | When first observed | In 2016, in France<br>81 problems verbatim | During manufacturing |
| HOW BIG/MANY | How many affected | • France—28 problems verbatim<br>• Germany—37 problems verbatim<br>• UK—16 problems verbatim | All vehicles built |
| | Defects per object | 3 | Multiple component failures |

All investigations made towards defining the problem will help the quality and engineering teams to identify the reasons for the failure and possible root causes.

## 4. Supporting Data

Starting from the need to respond to customer demands for compliant products, HVAC systems have a number of complaints registered against them regarding malfunctioning air

conditioning, heat loss, and noises. A number of optimal control tools are used to take into account the detected problems, the comfort of the cab, and the durability of the system [18].

The data obtained after defining the problem were customized and analyzed using the Pareto method. In Table 4, the identified data are sorted according to the nature of the problem and its recurrence. At the same time, a series of actions are listed that will be submitted in the conclusive verification of the identified problems.

**Table 4.** HVAC customer issues.

| HVAC Verbatims | Issues Count | Action |
| --- | --- | --- |
| Fan/blower—noisy | 20 | Check assembly issues. |
| Air conditioner—noisy | 16 | Check assembly issues |
| Windows fog up a lot | 12 | 1. Air does not blow on the windows<br>2. Air conditioner doesn't provide enough cold air |
| Other heating/cooling/ventilation problems | 11 | Check assembly issues |
| Air conditioner—does not get cold enough | 9 | Check AC compressor and Freon lines' pressure<br>Interior temperature sensor<br>Check if the fan is working properly |
| Air conditioner—not working properly | 4 | Check the overall issues |
| Heater/AC smells moldy/stale | 4 | Clean AC system and check |
| Cannot maintain desired temperature | 2 | Check AC compressor and Freon lines' pressure<br>Interior temperature sensor<br>Check if the fan working properly |
| Front defroster problem | 1 | Electrical issue—fuse or circuit broken |
| Rear defroster problem | 1 | Electrical issue—fuse or circuit broken |
| Controls do not work properly | 1 | Electrical issue—fuse or circuit broken |

The data and their recurrence are represented graphically and analyzed according to the Pareto method, and are shown in Figure 4.

According to the performed Pareto analysis, clients' concerns related to noises, non-functional air conditioning and heat loss were confirmed. Thus, three essential problems, fan/blower—noisy, air conditioner—noisy, and windows fog up a lot, are listed.

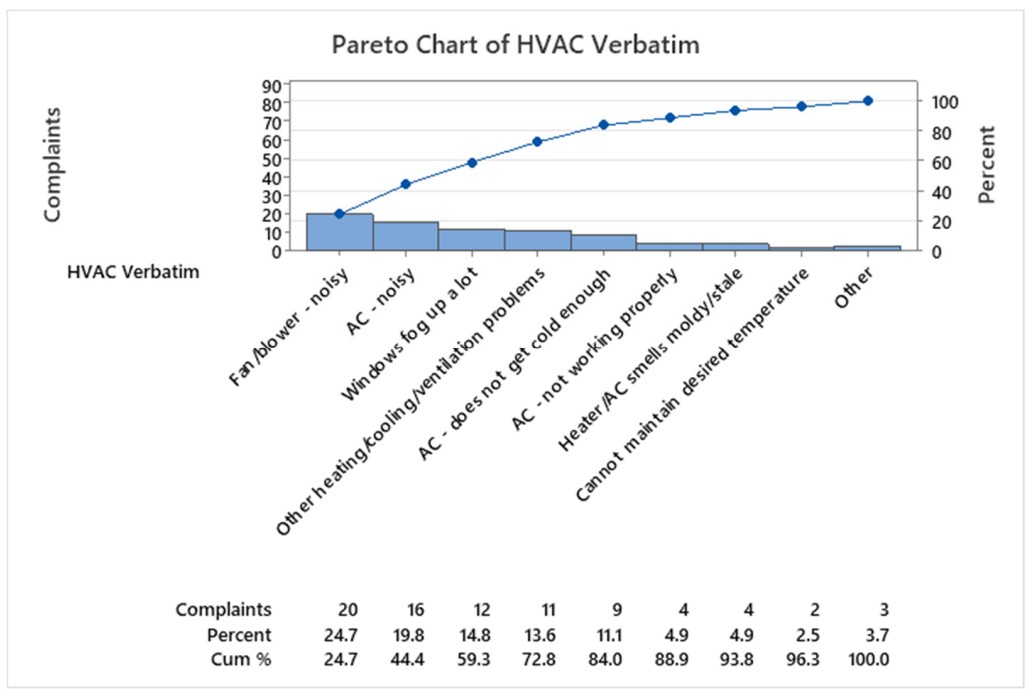

**Figure 4.** Pareto–HVAC issue analysis.

The conclusions of the Pareto analysis will help the engineering teams in implementing new approaches in an easier and more efficient way to solve the problems [19].

### 4.1. What Data Were Analyzed?

A series of data were analyzed in order to identify other adjacent issues, which are repaired by dealer services around the world. Many of the reasons for the repairs have not been identified and are unknown, as shown in Figure 5.

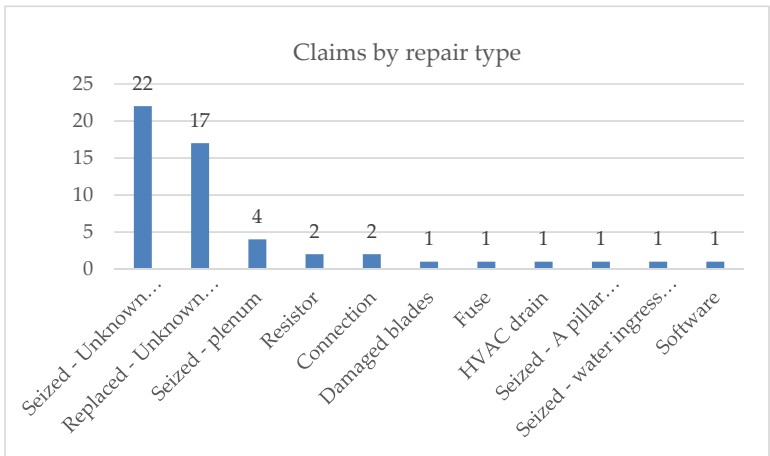

**Figure 5.** HVAC system repairs.

Repairs are made when the root cause is determined and a spare part will resolve the concern for the life of the vehicle.

In order to understand symptoms or to determine the failure mode and subsequent root cause, involved personnel may need to have the causal part returned for investigation. To achieve an understanding of customer issues, the teams review and process information from a number of systems.

### 4.2. Where Were the Problems Reported?

Based on the data recorded following customer complaints, an analysis was performed based on worldwide impact and where the highest incidence rates were recorded (Figure 6).

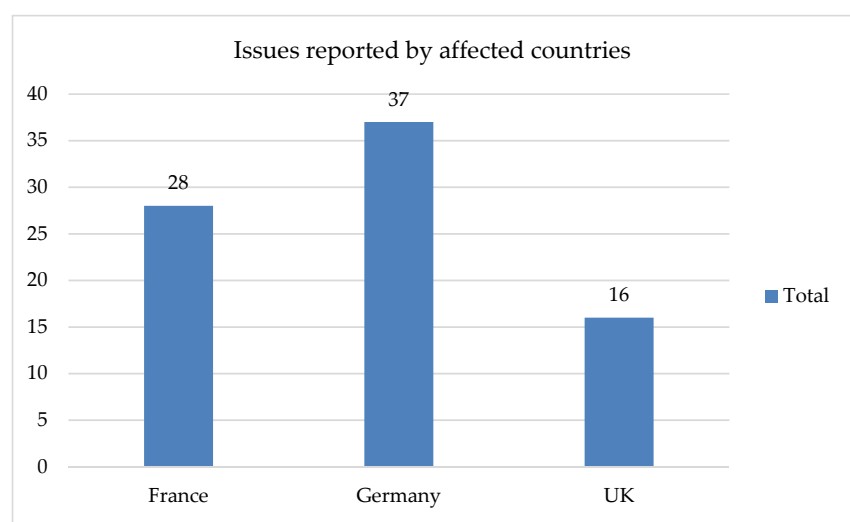

**Figure 6.** Issues reported by affected countries.

As can be seen in the graph, most complaints were registered in Germany, which had the highest degree of failures recorded.

### 4.3. How Big and How Many?

The graph in Figure 7 lists complaints as a probability graph and displays the matching of the specified distribution; data points are scattered along the normal distribution line.

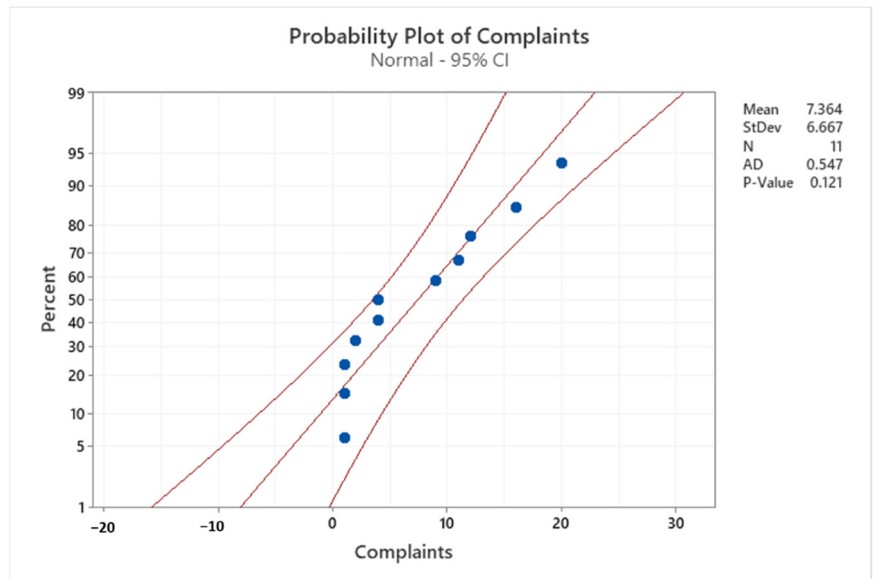

**Figure 7.** Plot of complaint representations.

The analysis in this paper has the aim of investigating and highlighting the input and output data that contributed to both the damage to the HVAC system and its repair. The data collected on the basis of customer complaints had an immediate impact on the company and the prioritization of improvement activities. The failure mode analysis presented at this stage of the paper found important clues that lead to the validation of possible solutions based on evidence.

## 5. Validation of the Study

In order to go through the entire cycle of validation of final conclusions following a performed analysis, the engineering and quality teams go through a detailed process of validation of theories in the manufacturing process, namely in the assembly line of the HVAC system, and these actions are carried out in accordance with the PCDA approach, as presented in Figure 8.

Based on this process of following the problem, planning actions, implementing possible solutions and validating them, the results of analyses, updating the work plan, reporting as well as sustainability result completion are communicated to teams and supervisors.

The data presented in this case study had an immediate effect in detecting problems and implementing immediate actions to find the main effects by checking each problem separately.

Following analyses, the following three main problems were identified:

1. Air conditioner/fan/blower—noisy;
2. Air conditioner—does not get cold enough;
3. Heater/AC smells moldy/stale.

A number of actions have been taken to identify the root causes. Actions were carried out in production, by visiting the retailer, and in test vehicles and climate chambers. All these actions were involved in the detection of root causes, as shown in Table 5.

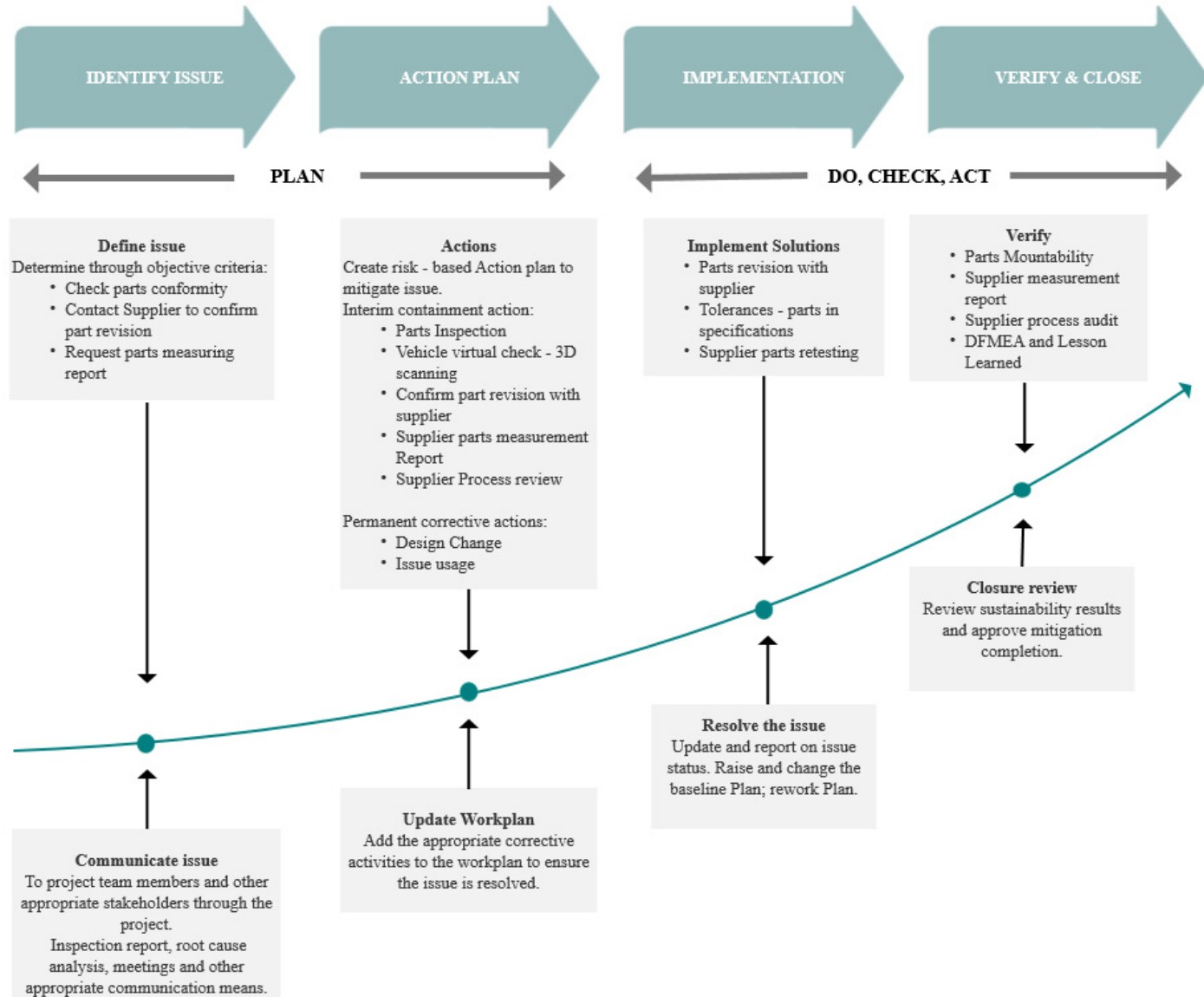

**Figure 8.** PDCA approach to problem solving.

A case study was analyzed and validated by applying the PDCA cycle for the continuous improvement of the quality of best practices and the impact on sustainability in the development of the strategy [20].

The PDCA approach [21] in this study helps to establish the main actions that should be considered and the implementation of the action plan that ensures best practices and sustainability [22,23]. The actions followed the PDCA approach in a logical course through the steps of planning the stages, double verification of the information and action, and applicability of the approaches for detecting the effects and causes. All these actions were carried out and are shown in Table 6, which presents the main root causes allocated to each detected problem.

Following the evolution of the results, it can be concluded that the root cause was blockage of the heater. All other assumptions will need to be verified to ensure failure prevention and customer satisfaction in an immediate response to their needs.

The results of the study confirm that HVAC systems failed to perform the necessary cooling and heating functions and meet appropriate standards [24].

**Table 5.** PDCA approach to sustainability.

| PDCA Cycle | PDCA Actions | Implementation | Best Practice | Sustainability Impact |
|---|---|---|---|---|
| Plan | Set goals and establish actions to identify defective AC components | Prepare action plan | Develop sustainability goals and identify stakeholders | Sustainability strategy development |
| Do | Provide a framework and information from dealership Visit retailer immediately | Build chart with findings (AC compressor, AC heater fan, AC pressure loss, etc.) | Improve action plan efficiency based on PCDA framework | Mitigate risks and capture opportunities |
| Check | Check for defective components and group them by criteria | Analyze the data and identify the failure mode | Determine the process KPIs to be measured and identify stakeholders to collect this data | ISO 13043:2011(en) certification |
| | Check for environmental issues by analyzing test reports Check for process issues by analyzing working procedures | Analyze the facts, replace the parts with supplier new parts. Warranty claim for repair | | Improve resource management and reduce environmental impact |
| ACT | Change assembly process Change procedure and work instructions | Fix the problem Implement solution | Test and monitor solution Plan and prioritize the projects | Streamline processes, reduce costs with findings, gain market share and satisfy customer concerns |
| | Set checkpoints to prevent recurrence | Regular quality check on components Periodic quality report from supplier Climatic hot/cold tests review for further projects Introduce validation plan for hot climate Introduce supplementary checks on assembly line | Repeat the tests to validate the test reports Quality control | Communicate sustainability efforts Lesson learned |

**Table 6.** Possible root causes.

| Identified Problems | Possible Root Cause |
|---|---|
| Air conditioner/fan/blower—noisy | Heater is blocked or the Freon levels are too low |
| Air conditioner—does not get cold enough | Heater is blocked or the Freon levels are too low |
| Heater/AC smells moldy/stale | Heater is blocked or pollen filter has not been replaced |

Quality processes and tools are provided to ensure that a customer has a relevant solution [25]. Lessons are also learned [26] and communicated so as not to reappear in future models, thus improving the quality of HVAC systems [27].

Closure occurs when the following criteria are met:

- Customer vehicle can be fixed at the retailer;
- Quality team informed that the closure is imminent in the interest of customers;
- A solution is available and has been demonstrated to be 100% effective;
- Software functionality has been validated and is available for download at retailers;
- Communication has been issued and engineering teams are satisfied that the fix is now available for them to start fixing customer cars;
- Preventing re-occurrence action is completed;
- Closure statement is created and submitted for acceptance and feedback rating.

## 6. Conclusions

During this study, the aim was to go through essential data of interest in detecting the main problems and to ensure the identification of causes that generated the non-conformities.

The first step was to define the issues through an initial quality study, a study performed in order to allocate scores depending on the nature of a problem and the vehicle

model. Data analysis was performed based on customer complaints, and the complaints were identified and collected from the French, German and British markets.

Defining problems helped this study in identifying the main three problems that had the greatest degree of impact on failure: HVAC is noisy, not cold enough, and has moldy smell issues. All these failure modes were detected using the 3Ws+H and the Pareto and PDCA methods.

An analysis of the root causes is shown via graphical representation of the data collected from customer complaints in world markets and by applying a series of verification actions.

The resulting actions were subjected to failure mode analyses using the PDCA cycle, an analysis that generates the implementation of actions, best practices, and sustainability impacts for current research.

The results showed that the main cause was that the heater being blocked.

The hypotheses described in this case study are subjected to the testing and validation process, to implement and improve solutions. To prevent the recurrence of failure, the data presented go through a final analysis and control process aimed at reducing the recurrence of defects in the HVAC vehicle segment. The quality control tools presented in the paper can be applied in the analysis of other components in the automotive industry or other sectors and to document actions for the improvement and identification of root causes and to provide the best solutions.

**Author Contributions:** Conceptualization, D.D. and I.S.; methodology, D.D.; validation, I.S.; formal analysis, D.D.; investigation, D.D.; resources, D.D., I.S. and N.D.I.; writing—original draft preparation, D.D.; writing—review and editing, D.D. and I.S.; visualization, D.D., I.S. and N.D.I.; supervision, I.S. and N.D.I. All authors have read and agreed to the published version of the manuscript.

**Funding:** Financial support from the Competitiveness Operational Program 2014–2020, Action 1.1.3: Creating synergies with RDI actions of the EU's HORIZON 2020 framework program and other international RDI programs, MySMIS Code 108792, Acronym project "UPB4H", financed by contract: 250/11.05.2020, is gratefully acknowledged.

**Institutional Review Board Statement:** Not applicable.

**Informed Consent Statement:** All collected data are anonymous, and subjects were informed about the full respect of data protection policies and ethics.

**Data Availability Statement:** Not applicable.

**Conflicts of Interest:** The authors declare no conflict of interest.

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
