# Peer review of "Quality Study on Vehicle Heat Ventilation and Air Conditioning Failure"

_sustainability, doi:10.3390/su132313441_

Round 1
Reviewer 1 Report
The paper is interesting, it deals with an important subject regarding vehicle heat ventilation and air conditioning failure, and it is my pleasure to review it.
However, I would have some considerations and suggestions for improving the quality of the article.
It is not clear how the analysed sample was selected, the indication "weighted by the volume of sales on the French, German and British markets" is insufficient, especially to ensure the representativeness (when, how was the weighting achieved, what were the criteria ?, what method was used?). Also, the characteristics of the sample are not well highlighted - age, obsolescence, nor the questions and answer options.
Chapter 4.1. What is an HVAC system? it is somehow placed too late in the text, after a lot of discussions and references made to this system. From our point of view, the subchapter is excessively technical, it fits more in a professional journal and not an academic one. We recommend limiting it to a few essential sentences, associated to the objective of the paper.
Referring to Table 5. Immediate action plan according to PDCA. Its content is somewhat inspired by technical textbooks or instructions and working protocols of mechanics, and seems quite simplistic in academic terms. Of course, its practical value may be indisputable, but an academic article has a more advanced role, to present results of scientific research, new knowledge or theories, interesting scientifically based and tested analyses of existing knowledge, etc.
The Conclusions do not seem to be … conclusions, but the first paragraph. Otherwise, the enumeration with the 5 bullets, and the subsequent sentences using systematically the future tense All hypotheses will be subject ...... The analysis will go through .... do not seem to summarize the present research. The limitations of the research and the orientation of the results within the main flow of the literature in the field (on similar topics) are not presented. We recommend a serious, careful restructuring of this chapter.
Formal
Improving the style, for example instead of According to C. S. Vishwanadha is recommendable According to Vishwanadha, and so on ... see Vranau etc ...
Fig.6. Issues reported, not repoted. ...., issue or issues?
The names of some tables and figures are schematic, without the necessary suggestiveness in the context, see Figure 6. Country’s evidence (whose? where? about what?) Is plural or singular (several countries appear in the figure).
Other minor editing issues - spaces, fonts, alignments
A general observation - taking into account the general orientation of the Journal - Sustainability - and the topic of this paper, we consider that more links could be made with sustainability (there were a lot of opportunities to do so).
Thank you for the opportunity to review this article and good luck!
Author Response
Dear Reviewer,
I sincerely thank you for reviewing my work and for the constructive advice on improving it.
At your recommendation, the improvements have been made and the paper Sustainability-1466558 can be found in the attachment.
- The data analysis was performed based on customer complaints, complaints identified and collected from the France, Germany and British markets. The 5Ws, Pareto, PDCA methods were used to interpret the results. I mention that the study was not conducted on the basis of samples, but on the basis of customer records. The corrections were inserted in the first part of the paper, in the introduction and state of art section, from page 1 to page 3.
- The subchapter on the HVAC system is indeed technical but reflects the main functional elements that contributed to the generation of the problems described in this study. We have limited the information to essential sentences, as recommended. As was mentioned, the description of the HVAC system was added later in the paper, so we added in the "state of art" stage its description to show the need for this study.
- The PDCA analysis table has been completely redone. As we were guided, the analysis reflects its scientific role in the presentation of knowledge and theories on the subject of the paper. PDCA analysis has added value in terms of sustainability and practices used to detect the need for improvement.
- The conclusions section of this paper has been redone and reorganized on page 14. The quality of the information on the records and results of the study has been improved.
- Following the observations made, the grammatical corrections were made.
- On the editor's advice, we presented the need for sustainability in the process of analyzing and improving the HVAC system.
Thank you again for the recommendations and I hope that the improvements will validate the paper work.
Reviewer 2 Report
This article is devoted to the description of the quality analysis performed by the authors in order to identify the main reasons for the mismatch of Vehicle Heat Ventilation and Air Conditioning (HVAC) systems with the expectations of consumers (Vehicle owners). The title of the article adequately reflects the content. In the abstract, the authors give the essence of the article, describe the state of the problem, research methods, results. Key words mostly correspond to the content of the article.
In the introduction, the authors provide a brief overview of research on the article topic, however, in our opinion, it would be more correct if the authors first indicated what is the purpose of determining customer satisfaction, as well as the quality control system's tasks for the automotive system in question. From this point of view, the structure of the literature review does not seem entirely logical to us, since customer satisfaction is a subjective indicator that depends on the target audience (more demanding customers; customers buying higher class vehicles; customers who service and operate the vehicle more carefully, etc.) At the same time, assessing the quality and identifying the causes of failures is a technical task, where objective assessments are applied and the culprit of the failure (designer, technologist, service worker or owner). The third section provides research on HVAS problems over a five-year period. However, it is not clear what activity type of this company has (due to the fact that these are different countries and different car brands, it is difficult to understand the activity type). At the section end, the authors write that "All studies conducted to determine the problem will help analyze the data obtained and identify possible causes." But it is not clear how they will help, since various methods are used for analysis, for each of which different types data with various properties are used. In the fourth section, the authors again try to analyze the data, but in the absence of a goal setting, it is not entirely clear who and what actions should be performed (either the vehicle owner or the auto mechanic). In addition, in our opinion, it would be logical to bring the structure of the system (HVAS) under consideration before analyzing the problems of its operation.
The fifth section, titled "Immediate Actions", presents a summary of the authors' findings from their analysis, which, in their opinion, will lead to "immediate effects." The authors provide a scheme that, in their opinion, will allow you to quickly identify the problem and eliminate it, which, in our opinion, is not entirely correct due to the above reasons. In conclusion, the authors list the research phases, and also indicate that all stages and hypotheses will be tested, but nevertheless, they write that this study can be extended to other parts and components of the vehicle, which, in principle, is incorrect, since for the system under consideration everything is based on owner subjective feelings, which is not applicable for many vehicle components in principle.
The article has been prepared in accordance with the instructions for authors, corresponds to the topic that it researches and publishes. But in our opinion, it is necessary to revise the structure of the article and name the sections and subsections more correctly. The list of references, not in our opinion, is insufficient. In our opinion, the article is consistent with the topic “vehicle quality from the point of view of the owners” and in type corresponds to the preliminary study.
Comment.
It is necessary to more clearly formulate the purpose of the study, as well as describe the methods used, the advantages of the proposed method over the works of other authors and its practical value. In our opinion, it is necessary to more clearly structure the text of the article in accordance with a more clearly formulated goal and objectives of the study.
There are comments on the design of the article (figures, tables and text):
- In the block diagram shown in Fig. 2, there is an ambiguity in the continuation of the process (three exits from logical blocks - for example, two times Yes. It is not clear what to do if the condition is met: return or continue)
- It is difficult to establish correspondence between different columns in tables (for example, in table 4: which records in column 3 refer to the numbers in column 2)
- There are typos in the text (for example, the numbering of subsections and references to tables in the text is broken) that need to be corrected.
Author Response
We kindly thank the Reviewer for corrections and improvement. As requested, corrections have been made and the paper Sustainability-1466558 can be found in the attachment.
- In the introductory part, on the editor's advice, was indicated and improved the purpose of determining customer satisfaction and the need for a quality control system was highlighted. At the same time, the emphasis is on determining the identification of the causes of the failure. The structure of the literature was based on the technical and scientific knowledge related by other authors. Emphasis was placed on the key issues encountered with the HVAC system, the possible problems that may arise, and the staging of possible causes, as presented in the state-of-the-art section on pages 2 and 3. The task was to identify failure mode in the most efficient way, in this case study.
- Case study: As requested, we presented the company's activity and the validity of the data collected. The study is conducted in a car company in the UK. The data presented were collected from worldwide markets, based on type of complaints from customers through a quality study presented in the paper. At this stage, renamed "Supporting data" - section 4, the data presented are obvious that support the analysis and validation of the study. All corrections have been made, as required.
- In section 5, renamed "Validation of the study", we highlighted the contribution brought by the analysis performed in the previous sections and its transposition in an efficient way using the PDCA approach the impact of sustainability in the efficiency of the study. The section, was corrected, as suggested.
- In the conclusions section, the context was redone and improved, in order to highlight the need for the case study and the contribution of the paper on identifying the root cause. The hypothesis was reformulated and it was indicated that quality tools can be used in other sectors or to analyze the failure of other components.
- The structure of the article has been revised and the subchapters renamed. The reference list has been revised, as advised.
- The purpose of the study was clearly reformulated, the case study structured and 5Ws, Pareto and PDCA methods presented.
- Following the comments:
- In the diagram in Figure 2, the Flowchart for the HVAC system process, if the condition is not met, returns to the HVAC system design to re-evaluate the possibilities and meet customer expectations.
- In Table 4, the information was restored and the correspondence between the columns was established.
- Typos in the text have been corrected.
Thank you again for the recommendations and I hope that the subject of the paper is more transparent and highlights the need for the present case study.
Round 2
Reviewer 1 Report
In the present version of the manuscript the authors addressed our suggestions in a correct and explicit manner. As a result, the paper is more suitable for publication.
Reviewer 2 Report
This article is devoted to the description of the quality analysis performed by the authors in order to identify the main reasons for the mismatch of Vehicle Heat Ventilation and Air Conditioning (HVAC) systems with the expectations of consumers (Vehicle owners). The title of the article adequately reflects the content. In the abstract, the authors give the essence of the article, describe the state of the problem, research methods, results. Key words correspond to the content of the article.
In the introduction, the authors provide a brief overview of the research on the topic of the article. The second section provides a diagram of the device and the sequence for testing it. The third section presents research on HVAS problems over a five-year period in the company that manufactures the system. The fourth section is devoted to data analysis.The fifth section, titled " Validation of the study", presents a summary of the authors' findings from their analysis, which, in their opinion, will lead to "immediate effects." The authors provide a scheme that, in their opinion, will allow you to quickly identify the problem and eliminate it. In conclusion, the authors present the results of the study.
The article has been prepared in accordance with the instructions for authors, corresponds to the topic that it researches and publishes. The list of references is sufficient. In our opinion, the article is consistent with the topic “vehicle quality from the point of view of the owners” and in type corresponds to the preliminary study.